

# Thermoelectric coefficients and the figure of merit for large open quantum dots

**Robert S. Whitney[1⋆] and Keiji Saito[2]**

**1** Laboratoire de Physique et Modélisation des Milieux Condensés,
Université Grenoble Alpes and CNRS, BP 166, 38042 Grenoble, France.
**2** Department of Physics, Keio University, Yokohama 223-8522, Japan.

⋆ robert.whitney@grenoble.cnrs.fr

## Abstract

We consider the thermoelectric response of chaotic or disordered quantum dots in the limit of phase-coherent transport, statistically described by random matrix theory. We calculate the full distribution of the thermoelectric coefficients (Seebeck $S$ and Peltier $\Pi$), and the thermoelectric figure of merit $ZT$, for large open dots at arbitrary temperature and external magnetic field, when the number of modes in the left and right leads ($N_{\text{L}}$ and $N_{\text{R}}$) are large. Our results show that the thermoelectric coefficients and $ZT$ are maximal when the temperature is half the Thouless energy, and the magnetic field is negligible. They remain small, even at their maximum, but they exhibit a type of universality at all temperatures, in which they do not depend on the asymmetry between the left and right leads ($N_{\text{L}} - N_{\text{R}}$), even though they depend on ($N_{\text{L}} + N_{\text{R}}$).



# 1 Introduction

There has long been interest in the thermal and thermoelectric response of nanostructures [1–5]. However, research in this field was restricted by the absence of good thermometry techniques at the nanoscale, which made it extremely hard to quantify heat flows. In recent years, numerous such thermometry techniques have been developed, leading to a renewed interest in thermal transport and thermoelectric effects at the nanoscale, see for example Refs. [6,7] and references therein.

In many cases, the objective is to maximize the thermoelectric response for applications, such as efficient new power sources or refrigerators, or for the recovery of useful power from waste heat. However, at a more fundamental level, the measurement of a thermoelectric response is a very interesting probe of nanostructures. It always gives us different information from the measurement of the nanostructure's electrical conductance. At a hand waving level, one can say that the thermoelectric response tells us about the difference between the dynamics of charge carriers above and below the chemical potential, when the electrical conductance only tell us about the sum of the two [6]. It is important to develop good quantitative models of the thermoelectric response of all kinds of nanostructures, to help us use the thermoelectric response as a quantitative probe of a nanostructures. In this context, we should maintain our interest in modelling nanoscale systems whose thermoelectric response is small, as well as in modelling those whose response is large. Large quantum dots are systems with a small but interesting thermoelectric response, which we study here.

Disordered or chaotic quantum dots have random properties whose statistics are described by random matrix theory. In particular, such dots have a spread of Seebeck coefficients, $S$, centred on zero, which means the average of the Seebeck coefficient over realizations of the disorder or the chaos, $\langle S \rangle$, is zero. The Peltier coefficient in such a two-terminal dot is $\Pi = T S$ for arbitrary external magnetic field [3][1], so it is also zero on average. However, in such a situation, if one takes a single typical dot, it will have Seebeck and Peltier coefficients of random sign but of non-zero magnitude. The typical value of such coefficients will be

$$S_{\text{typical}} \simeq \pm \sqrt{\langle S^2 \rangle} \quad \text{with} \quad \Pi_{\text{typical}} = T S_{\text{typical}}. \tag{1}$$

The dimensionless figure of merit $ZT$ which measures the system's efficiency as a thermoelectric heat engine (see table 1 in Ref. [6]), or its efficiency as a refrigerator, is

$$ZT = \frac{GS^2T}{K}, \tag{2}$$

---

[1]The Onsager relation for a two terminal system is $\Pi(B) = T S(-B)$, where $B$ is the external magnetic field. However, Ref. [3] pointed out that a phase coherent nanostructure (in which electrons undergo no inelastic scattering or Andreev reflection) must have $S(B) = S(-B)$; so then $\Pi(B) = T S(B)$ for all $B$.

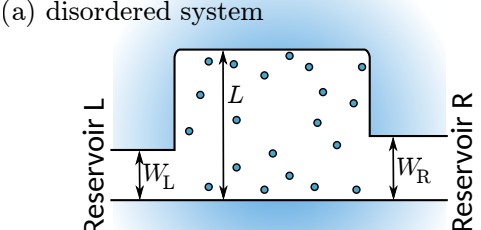
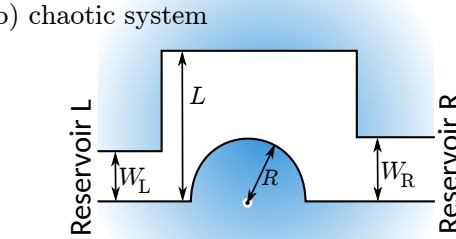

Figure 1: The systems we consider in this work are those described by random matrix theory, this could be (a) a disordered system or (b) a chaotic system (such as a Sinai billiard). In both cases, the system is coupled to two reservoirs (left and right) through leads with $N_\mathrm{L}$ and $N_\mathrm{R}$ modes respectively. The number of lead modes $N_i$ for a lead of width $W_i$ is given above Eq. (12b). We define $N = N_\mathrm{L} + N_\mathrm{R}$, and consider the properties of the ensemble of such systems for $N \gg 1$, as modelled by an ensemble of random matrices. In the case of the disordered system, this ensemble corresponds to the ensemble of systems with the disorder in different positions. In the case of chaotic systems, this ensemble corresponds to an ensemble of shapes of the potential; for the Sinai billiard, such an ensemble can be found by varying the radius $R$ over a range much larger than the particle wavelength but much smaller than the system size.

for temperature $T$, electrical conductivity $G$ and thermal conductivity $K$. Since this contains $S^2$, and we know both $G$ and $K$ must be positive, we see that this will not average to zero. It would be useful to know both $S_\mathrm{typical}$ and $\langle ZT \rangle$ as a function of system parameters and temperature. Even better would be to have the full distribution of $ZT$, so one could answer questions such as, what percentage of samples will have a $ZT$ greater than a given value.

This work treats such a problem in the simplest case, spinless electrons flowing through a large open quantum dots, where the dot is coupled to many reservoir modes, $N \gg 1$. This implies that the level broadening in the dot is of order $N$ times the dot's level spacing, so the transmission though the dot is only weakly energy dependent. Since high efficiencies rely on good energy filtering, it should be no surprise that this limit has a $ZT$ much less than one.

The seminal work on this subject was that of van Langen, Silvestrov, and Beenakker [8], which contains a plethora of results on the Seebeck coefficient in coherent transport through disorder wires and dots. Here, we revisit their result for large $N$, stated in the conclusions of Ref. [8], which says that the typical magnitude of the Seebeck coefficient $S_\mathrm{typical}$ is small (in units of $k_\mathrm{B}/e$) but grows with increasing temperature. It is clear that the result was only for the low temperature limit (temperature much smaller than the Thouless energy), and it is not hard to guess that $S_\mathrm{typical}$ will not continue to grow with temperature for larger temperatures. However, it raises the question; what value of temperature maximizes the thermoelectric response, and how big is this thermoelectric response?

This work answers this question, making the following modest but four-fold contribution to the subject.

(i) We extend the result in Ref. [8] to arbitrary temperatures and arbitrary magnetic fields (although as we only treat spinless electrons, spin-orbit coupling is not present). This shows that the thermoelectric coefficients (Seebeck and Peltier effects) are maximized when $k_\mathrm{B}T$ is very close to half the Thouless energy, $E_\mathrm{Th}$, and the external magnetic field is negligible. At this peak, the thermoelectric coefficients scale like $N^{-2}$ for large $N$, which is very different from the scaling like $N^{-4}$ found at low temperatures in Refs. [8, 11].

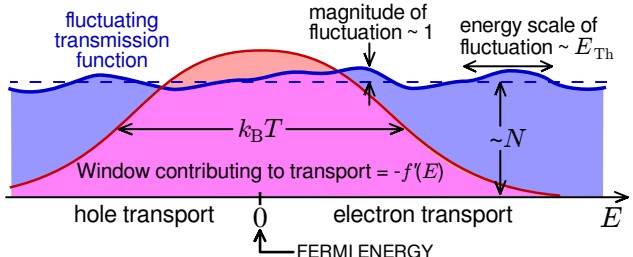

Figure 2: A sketch of the transmission of a a typical disordered or chaotic dot for large $N$. The transmission (blue curve) is of order $N$ on average, but with fluctuations of order one, that is to say of order $N^0$. The factor of $-f'(E)$ (red curve) is that given in Eq. (5). The transport properties are related to the product of these two functions, so that they are dominated by the transmission in an energy window of order $k_B T$ around the Fermi energy.

(ii) This maximum in the Seebeck coefficient also corresponds to the maximal thermoelectric figure of merit, $ZT$. Not unexpectedly, it remains much smaller than one even at its maximum, see section 1.1 below.

(iii) We recover Ref. [8]'s result that the distribution of $S$ is Gaussian, and show that this implies that the distribution of $ZT$ is an exponentially decaying function of $ZT$, with a square-root divergence at small $ZT$.

(iv) We show that both $S$ and $ZT$ have a type of universality, in the sense that they do not depend on the asymmetry in the number of modes in the left and right leads (even though they depend on the total number of lead modes, $N$). For example, $S$ and $ZT$ have exactly the same average value and exactly the same distribution for a dot with highly asymmetric leads with $N_L = 100$ and $N_R = 300$, as for a dot with symmetric leads with $N_L = N_R = 200$.

In all cases, we explain how to derive these results (the results in Ref. [8] were mainly stated without derivation) using a diagrammatic method that works for chaotic or disordered systems that obey random matrix theory.

## 1.1 Why small thermoelectric coefficients and small $ZT$?

In non-interacting systems the only way to have a thermoelectric effect is to have very different dynamics of particles above and below the Fermi energy. The best thermoelectric is thus a system that acts as an energy filter, letting particles flow between hot and cold at certain energies, and blocking their flow at other energies [6].

In contrast, the systems considered here (with $N \gg 1$) let particles flow at all energies, so one can guess their thermoelectric response will be small. This can be understood intuitively by thinking of the dot's discrete spectrum of randomly placed levels with a mean level spacing of $\Delta$. The coupling to the leads broadens these levels[2] on the scale of $E_{Th} \sim N\Delta$. This means that for $N \gg 1$, the broadening is much greater than the spacing between levels, so the dot's density of states is rather flat, with small fluctuations on the scale of $E_{Th}$. This implies the transmission as a function of energy is also rather flat, with small fluctuations on the scale of $E_{Th}$, see Fig. 2. Temperature determines a window of energies, $k_B T$, on which transmission contributes to $S^2$ and $ZT$. If this window is much less than $E_{Th}$, then the transmission is

---

[2]Of course, this is only a hand waving argument to see the energy scales. In reality, adding large leads to a isolated quantum system so strongly perturbs the system that there is no clear relation between the energy levels of the isolated system and the density of states of the the system with leads.

almost energy independent, and $ZT$ vanishes. If this window is much more than $E_{\mathrm{Th}}$, then the transmission's oscillations are on a much smaller scale than the window; They thus tend to cancel each other, which again means that $S^2$ and $ZT$ vanishes. If this window is of order $E_{\mathrm{Th}}$, then the transport is at its most sensitive to the fluctuations in transmission, so both $S^2$ and $ZT$ will be maximal.

Quantitative studies of conductance fluctuations have shown that these fluctuations in transmission have a magnitude of order one (when the average transmission is of order $N$), and do indeed fluctuate on the scale of $E_{\mathrm{Th}}$, see Fig. 2. Given this information, readers already familiar with the scattering theory for thermoelectric effects will probably be able to guess that our calculations will show that the Seebeck coefficient at its peak is typically of order $1/N$ in units of $e/h$, and $ZT$ of order $1/N^2$. However, as the position of these fluctuations are random, some samples will have small positive Seebeck coefficient (i.e. slightly higher transmission for particles above the Fermi surface than below the Fermi surface), while others will have have a small negative Seebeck coefficient (i.e. transmitting particles below the Fermi energy a bit more than those above). Hence, $\langle S \rangle = 0$, while $\langle S^2 \rangle$ is finite but small, where small means $\langle S^2 \rangle \sim (k_{\mathrm{B}}/e)^2 N^{-2}$ for large $N$. By the same arguments, the dimensionless figure of merit is small, with $\langle ZT \rangle \sim N^{-2}$ for large $N$.

The remaining questions are all quantitative ones; what is the exact temperature dependence of $S$ and $ZT$. Where exactly in the peak, how big is it, how does it depend on magnetic field, etc?

## 1.2 Works on similar systems in different regimes

Ref. [8] discussed the low temperature limit for both the limit of large $N$ and small $N$, with Refs. [9, 10] going further with small $N$. Ref. [11] extended these low temperature results to systems with decoherence and relaxation, as modelled by a voltage probe. Ref. [12] is another interesting recent work for small $N$, which considers the band-edge of a system described by a random matrix, where the levels are very sparse, so can be modelled by just two such levels, and considers the limit where the thermal conductance is dominated by phonons, so $ZT = GS^2 T/K_{\mathrm{phonon}}$, and the statistics of $ZT$ are entirely determined by the statistics of $GS^2$. Another work which considered open quantum dots (in the large $N$ limit) is Ref. [13], however this mainly considers the non-zero value of $\langle S \rangle$ that is induced by a superconducting loop threaded by a flux.

While it is not the subject of this work, it is worth mentioning a significant activity on the thermoelectric response of disordered one-dimensional wires, both in the coherent transport regime [8, 14–16], and the variable range hopping regime [15, 17–20].

This work concentrates on large open quantum dots, which have small thermoelectric response for the reasons outlined above, however there are many works on smaller more closed nanostructures, which exhibit strong thermoelectric responses. Examples include a quantum point-contact or a quantum dot with weak tunnel-coupling to the reservoirs, for a review see sections 4-6 of Ref. [6]. Similar nanoscale systems with interactions were reviewed in section 7-9 of Ref. [6]. For the large open systems considered here, we neglect interactions; however there has long been evidence that a thermoelectric response can be slightly modified by interactions, see e.g. [21, 22]. We do not consider such interaction effects here.

## 2 Thermoelectric coefficients and figure of merit

We consider systems of the types shown in Fig. 1 in the linear response regime. Refs [1–3, 5] showed that the thermoelectric transport through such systems can be written in terms of integrals over the transmission function $\mathcal{T}(E)$ via the Landauer scattering theory [23, 24]; for

a review see Chapters 4 and 5 of Ref. [6]. One can write the electrical conductance $G$, thermal conductance $K$. Seebeck coefficient $S$, and Peltier coefficient $\Pi$ in terms of coefficients of the Onsager matrix, which can then be written in term of integrals over the transmission function (see e.g. section 5.2 of Ref. [6]). Then

$$G = e^2 I_0, \qquad K = \frac{1}{T}\left(I_2 - \frac{I_1^2}{I_0}\right), \tag{3}$$

$$S = \frac{1}{eT}\frac{I_1}{I_0}, \qquad \Pi = \frac{1}{e}\frac{I_1}{I_0}, \tag{4}$$

with the integral

$$I_n \equiv \int_{-\infty}^{\infty} \frac{dE}{h}\, (E-\mu)^n\, \mathcal{T}(E,\boldsymbol{B})\big[-f'(E)\big], \tag{5a}$$

where $\mu$ is the chemical potential, and $f'(E)$ is the derivation of the Fermi function with respect to energy, so

$$f'(E) = -\frac{1}{4k_{\mathrm{B}}T\cosh^2\big[E/(2k_{\mathrm{B}}T)\big]}. \tag{5b}$$

The Seebeck coefficient of such a dot is zero on average because its sign is random, however a typical system will have a Seebeck coefficient of order $\pm\sqrt{\langle S^2\rangle}$. Ref. [8] already gave a result for $\langle S^2\rangle$ in the large $N$ limit (see the Conclusions at the end of Ref. [8]), which grows like $(k_{\mathrm{B}}T/E_{\mathrm{Th}})^2$. It is natural to guess that this result can only hold for small $T$, and the quadratic growth must stop at some value of $T$. Here we ask, what is the $T$-dependence at larger $T$, and how big can the typical Seebeck coefficient become? At the same time, we ask what is the average value of the figure of merit $ZT$ as a function of temperature? For $ZT$, combining Eqs. (2-4), one finds that

$$ZT = \frac{I_1^2}{I_2 I_0 - I_1^2}. \tag{6}$$

One immediately sees from Eqs. (4,6) that to average $S^2$ or $ZT$ over realizations of the chaos or disorder, one is required to take averages of products and ratios of integrals containing transmission functions. In general, this is a highly difficult technical problem. However, in the limit of $N \gg 1$, the situation is greatly simplified by the fact that $I_0$ and $I_2$ have fluctuations much smaller than their average. Writing $I_0 = \langle I_0\rangle + \delta I_0$ and $I_2 = \langle I_2\rangle + \delta I_2$, and counting powers of $N$ in the manner described later in this article (or in Ref. [25]), one finds that $\langle I_0\rangle \sim \langle I_2\rangle \sim N$ while $I_1 \sim \delta I_0 \sim \delta I_2 \sim 1$ [3]. Thus

$$\langle S^2\rangle = \frac{1}{e^2 T^2}\frac{\langle I_1^2\rangle}{\langle I_0\rangle^2}\big[1+\mathcal{O}(1/N)\big], \tag{7}$$

$$\langle ZT\rangle = \frac{\langle I_1^2\rangle}{\langle I_2\rangle\,\langle I_0\rangle}\big[1+\mathcal{O}(1/N)\big]. \tag{8}$$

## 2.1 Evaluating the averages in Eqs. (7,8)

We will treat the problem of finding averages of $I_0$, $I_1$ and $I_2$ in the limit of $N \gg 1$ using the diagrammatic rules, which were derived from a semiclassical treatment of trajectories in a chaotic cavity, and shown to coincide exactly with the results of random matrix theory [26–28].

---

[3]This argument also allows one to arrive at the well known results that the conductances, $G$ and $K$, are well approximated by their average values, $\langle G\rangle$ and $\langle K\rangle$ for large $N$, see section 2.2.

This work will use two results of this diagrammatic method derived in Refs. [28–33]. The first result, whose derivation we outline in Section 5.2 is

$$\left\langle \mathcal{T}(E) \right\rangle \;\; = \;\; \frac{N_\mathrm{L} N_\mathrm{R}}{N} \left[ 1 + \mathcal{O}(1/N) \right], \tag{9}$$

where $N = N_\mathrm{L} + N_\mathrm{R}$ is the total number of lead modes. Defining the deviation of the transmission from its average value as,

$$\delta \mathcal{T}(E) = \mathcal{T}(E) - \left\langle \mathcal{T}(E) \right\rangle, \tag{10}$$

the second result of the diagrammatic method in Section 5.2 is

$$\left\langle \delta \mathcal{T}(E_1) \delta \mathcal{T}(E_2) \right\rangle = \frac{N_\mathrm{L}^2 N_\mathrm{R}^2}{N^4} \left\{ \left( 1 + \frac{B^2}{B_\mathrm{c}^2} + \frac{\Delta E^2}{E_\mathrm{Th}^2} \right)^{-1} + \left( 1 + \frac{\Delta E^2}{E_\mathrm{Th}^2} \right)^{-1} \right\} \left[ 1 + \mathcal{O}\!\left( \frac{1}{N} \right) \right], \tag{11}$$

where for compactness we define $\Delta E = E_2 - E_1$. The two parameters in $\left\langle \delta \mathcal{T}(E_1) \delta \mathcal{T}(E_2) \right\rangle$ are the Thouless energy, $E_\mathrm{Th}$, and the crossover field, $B_\mathrm{c}$. The Thouless energy is

$$E_\mathrm{Th} = \hbar / \tau_\mathrm{D}, \tag{12a}$$

with $\tau_\mathrm{D}$ being the average time a particle spends in the dot. For a $d$-dimensional dot of typical volume $L^d$, and with lead $i$ having width $W_i$ and so carrying $N_i \sim (p_\mathrm{F} W_i / h)^{d-1}$ modes, one has

$$E_\mathrm{Th} \sim \frac{\hbar v_\mathrm{F}}{L^d} \left( W_\mathrm{L}^{d-1} + W_\mathrm{R}^{d-1} \right) \sim \frac{\hbar v_\mathrm{F} N}{L} \left[ \frac{h}{p_\mathrm{F} L} \right]^{d-1}, \tag{12b}$$

where $v_\mathrm{F} = p_\mathrm{F}/m$ is the Fermi velocity and we drop many factors of order one. This Thouless energy can be thought of as the broadening of dot levels due to the finite probability of escape into the leads; it is of order $N$ times the dot level-spacing. Thus for large $N$, the dot has a smooth and almost constant density of state, which means that the dot's transmission function is also smooth and almost constant (see Fig. 2). The crossover field, $B_\mathrm{c}$, is the external magnetic field which induces the crossover from the time-reversal symmetric system to the one with broken time-reversal symmetry. Physically, $B = B_c$ corresponds to the field where a typical trajectory through the system encloses one flux quantum. For a system of area $A \sim L^2$, which an electron with velocity $v_\mathrm{F}$ crosses in a time $\tau_0 \sim L/v_\mathrm{F}$, one has

$$B_c \sim (h/eA)(\tau_0/\tau_\mathrm{D})^{1/2}, \tag{13}$$

see e.g. Section VE of Ref. [34] or Section IIB4 of Ref. [25].

Since $\left\langle \mathcal{T}(E) \right\rangle$ is energy independent, the integrals of it over $E$ are known for $n = 0, 1, 2$. Hence,

$$\left\langle I_n \right\rangle = \frac{1}{h} \frac{N_\mathrm{L} N_\mathrm{R}}{N} \times \begin{cases} 1 & \text{for} \quad n = 0, \\ 0 & \text{for} \quad n = 1, \\ \pi^2 k_\mathrm{B}^2 T^2 / 3 & \text{for} \quad n = 2, \end{cases} \tag{14}$$

at leading order in $1/N$. Next, we note that

$$\left\langle I_1^2 \right\rangle \;\; = \;\; \int dE_1 dE_2 \, \frac{(E_1 - \mu)(E_2 - \mu)}{h^2} \left\langle \mathcal{T}(E_1) \mathcal{T}(E_2) \right\rangle f'(E_1) f'(E_2), \tag{15}$$

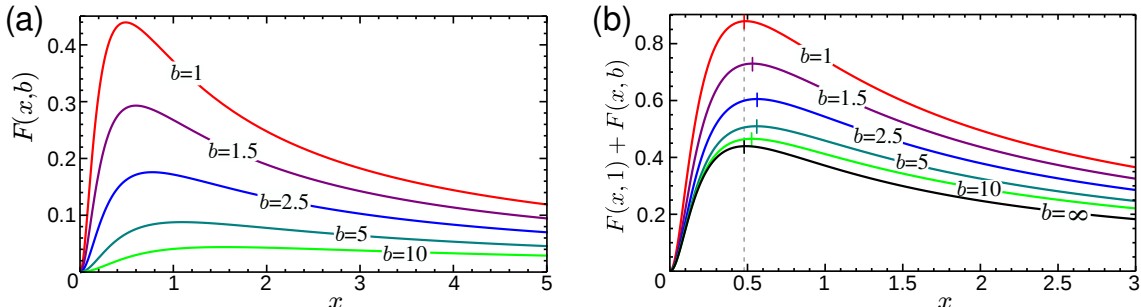

Figure 3: (a) A plot of $F(x, b)$ given by the integral in Eq. (16b), for various $b$ (we are only interested in $b \geq 1$). The function goes like $x^2$ for small $x$ (on a scale almost too small to see here), while it goes like $1/x$ for large $x$. (b) A plot of $F(x, 1) + F(x, b)$ which appears in $\langle S^2 \rangle$ in Eq. (23) and $\langle ZT \rangle$ in Eq. (27), with the vertical dash marking the maximum, this maximum is at $x \simeq 0.48$ for $b = 1$ and $b = \infty$, but at slightly larger $x$ for intermediate values of $b$.

so using Eq. (11) and defining $y_i = (E_i - \mu)/(k_B T)$, we find that

$$\langle I_1^2 \rangle = \frac{N_L^2 N_R^2}{N^4} \frac{k_B^2 T^2}{h^2} \left[ F\left( \frac{k_B T}{E_{Th}}, 1 \right) + F\left( \frac{k_B T}{E_{Th}}, 1 + \frac{B^2}{B_c^2} \right) \right], \tag{16a}$$

where to lowest order in $1/N$,

$$F(x, b) = \frac{1}{16} \int \frac{dy_1 dy_2 \, y_1 y_2}{b + (y_1 - y_2)^2 x^2} \frac{1}{\cosh^2[y_1/2] \cosh^2[y_2/2]}. \tag{16b}$$

This function is plotted in Fig. 3a, it is non-monotonic with a maximum when $b = 1$ and $x = 0.48 \cdots$ (given that we are only interested in $b \geq 1$). This tells us that the maximum $\langle I_1^2 \rangle$ occurs when $k_B T/E_{Th} \simeq 0.48$ and $B \ll B_c$.

To get the small $x$ behaviour of $F(x, b)$ analytically, we expand $[b + (y_1 - y_2)^2 x^2]^{-1} = 1/b - (y_1 - y_2)^2 x^2/b^2 + \cdots$ in the integrand. The leading order in $F(x, b)$ is then $F(0, b) = 0$, because the integrand is an odd function of $y_1$ and $y_2$, hence the integral is dominated by the order $x^2$ term at small $x$. Evaluating the integral over $y_1, y_2$ in the $x^2$ term, we find for small $x$ that

$$F(x, b) = \frac{2\pi^4 x^2}{9 \, b^2} \times \left( 1 + \mathcal{O}[x^2/b] \right). \tag{17}$$

To get the large $x$ behaviour of $F(x, b)$, one can use the hand-waving argument that the integrand is of order $y_1 y_2$ in a rectangle given by $-2 < (y_1 + y_2) < 2$ and $-1/x < y_1 - y_2 < 1/x$, and is small enough to neglect outside this rectangle. One then goes to coordinates $z_{\pm} = y_1 \pm y_2$, so $y_1 y_2 \sim z_+^2 - z_-^2$, for which the integral takes the form $\int_{-2}^{2} dz_+ \int_{-1/x}^{1/x} dz_- \left( z_+^2 - z_-^2 \right) \sim 1/x$ (dropping all prefactors of order one).

## 2.2 Conductances and the Wiedemann-Franz ratio

Before addressing thermoelectric effects we mention the electrical conductance, $G$, and thermal conductances, $K$. Given Eqs. (3,14), we immediately arrive at the well-known result for

the average conductance

$$\langle G \rangle \quad = \quad e^2 \langle I_0 \rangle = \frac{e^2}{h} \frac{N_L N_R}{N} , \tag{18}$$

at leading order in $1/N$. Using the same arguments as given above Eq. (7), we see that the second term in $K$ in Eq. (3) does not contribute to $\langle K \rangle$ at leading order in $1/N$. Hence

$$\langle K \rangle \quad = \quad \frac{1}{T} \langle I_2 \rangle = \frac{\pi^2 k_B^2 T}{h} \frac{N_L N_R}{N} , \tag{19}$$

at leading order in $1/N$. Furthermore, the arguments given above Eq. (7) convince us that any such system will only have small deviations in its conductances from this average value, so $G = \langle G \rangle \big[ 1 + \mathcal{O}(1/N) \big]$ and $K = \langle K \rangle \big[ 1 + \mathcal{O}(1/N) \big]$. Hence the system satisfies the Wiedemann-Franz law [5],

$$\frac{K}{G} = \frac{\langle K \rangle}{\langle G \rangle} = \frac{1}{e^2 T} \frac{\langle I_2 \rangle}{\langle I_0 \rangle} = \mathcal{L} T , \tag{20}$$

to leading order in $1/N$, where $\mathcal{L}$ is the Lorenz number

$$\mathcal{L} = \frac{\pi^2 k_B^2}{3 e^2} . \tag{21}$$

Ref. [5] points out that while this holds to leading order in $1/N$, there are mesoscopic fluctuations which violate the Wiedemann-Franz law at the next order.

When Eq. (20) holds, we see immediately that $ZT$ and $S$ are directly related via

$$ZT \quad = \quad S^2 \big/ \mathcal{L} . \tag{22}$$

Hence this holds for the systems considered here to leading order in $1/N$.

## 2.3   Typical magnitude of the thermoelectric effects

The Seebeck coefficient of such a dot is zero on average because its sign is random, however a typical dot will have a Seebeck coefficient given in Eq. (1). Ref. [8] already gave a result for $\langle S^2 \rangle$ in the large $N$ limit (see the Conclusions at the end of Ref. [8]), which grows like $(k_B T / E_{Th})^2$. It is natural to guess that this result can only hold for small $T$, and the quadratic growth must stop at some value of $T$. Here we ask, what is the $T$-dependence for arbitrary $T$, and how big can the typical Seebeck coefficient become?

Substituting the averages in Section 2.1 into Eq. (7), we find that

$$\langle S^2 \rangle \quad = \quad \frac{k_B^2}{e^2} \frac{1}{N^2} \left[ F \left( \frac{k_B T}{E_{Th}}, 1 \right) + F \left( \frac{k_B T}{E_{Th}}, 1 + \frac{B^2}{B_c^2} \right) \right], \tag{23}$$

to leading order in $1/N$. Of course, as $\Pi = TS$ in these systems, the average of the square of the Peltier coefficient is simply

$$\langle \Pi^2 \rangle \quad = \quad T^2 \langle S^2 \rangle . \tag{24}$$

Thus, the typical quantum dot will have a non-zero Seebeck and Peltier coefficients given by substituting Eq. (23) into Eqs. (1).

In the small temperature limit, this result coincides with that in the conclusions of Ref. [8]. To see this, one use Eq. (17) and take the fact that $E_{Th} \sim N \Delta$ as mentioned below Eq. (12b), so

$$\langle S^2 \rangle \quad = \frac{k_B^2}{e^2} \frac{k_B^2 T^2}{N^4 \Delta^2} \qquad \text{for } k_B T \ll N \Delta \tag{25}$$

as in Refs. [8, 11], for a level spacing of $\Delta$. An attentive reader will note that the numerical value of the constant of proportionality is a little different here from in Refs. [8, 11], because of an ambiguity of order one in the choice of definition of $E_{\mathrm{Th}}$, see our Eq. (12b).

The crucial points are that the quadratic growth of $\langle S^2 \rangle$ with $T$ is only for extremely small $T$ (so small it is barely visible in the plot in Fig. 3b), and that there is a peak in $\langle S^2 \rangle$ at $k_{\mathrm{B}}T/E_{\mathrm{Th}} \simeq 0.48$ and $B \ll B_{\mathrm{C}}$. The maximum value, given by this peak, is

$$\langle S^2 \rangle_{\max} \simeq 0.88 \times \left( \frac{1}{N} \frac{k_{\mathrm{B}}}{e} \right)^2. \tag{26}$$

The full non-linear dependence on $k_{\mathrm{B}}T/E_{\mathrm{Th}}$ shown in Fig. 3b, combined with the fact that $E_{\mathrm{Th}} \sim N\Delta$, means that the dependence of $\langle S^2 \rangle$ on both $T$ and $N$ is non-linear. Broadly speaking there are four regimes of behaviour, in order of increasing temperature,

- At very small $T$ [8,11], we have $\langle S^2 \rangle \sim N^{-4}T^2$. This regime with $T^2$ behaviour is almost too small to see in the plot in Fig. 3b.

- There is a regime of approximately linear $T$-dependence visible in Fig. 3b, created by the cross-over from the quadratic small $T$-dependence to the peak at $k_{\mathrm{B}}T \sim E_{\mathrm{Th}}/2$. In this regime[4], $\langle S^2 \rangle$ goes like $N^{-3}T$.

- The regime of the peak ($k_{\mathrm{B}}T \sim E_{\mathrm{Th}}/2$) has $\langle S^2 \rangle$ going like $N^{-2}$.

- The regime where $k_{\mathrm{B}}T \gg E_{\mathrm{Th}}/2$, has $\langle S^2 \rangle$ which decays with increasing $T$, going like $1/(NT)$.

The different dependences on $N$ in these four different temperature regimes should be observable in those experiments where one can easily vary the lead width with split gates.

For large magnetic fields, $B \gg B_{\mathrm{c}}$, the value of $\langle S^2 \rangle$ at any $T$ is half that with $B = 0$ at the same $T$.

## 2.4 Average figure of merit

To get the average of the dimensionless figure of merit, $ZT$, we can substitute the averages in Section 2.1 into Eq. (8), or we can simply substitute Eq. (23) into Eq. (22). In either case we find that

$$\langle ZT \rangle \;\; = \;\; \frac{3}{\pi^2 N^2} \left[ F\left( \frac{k_{\mathrm{B}}T}{E_{\mathrm{Th}}}, 1 \right) + F\left( \frac{k_{\mathrm{B}}T}{E_{\mathrm{Th}}}, 1 + \frac{B^2}{B_{\mathrm{c}}^2} \right) \right]. \tag{27}$$

The conditions that maximize $\langle ZT \rangle$ are the same as those that maximize $\langle S^2 \rangle$ (from the curves in Fig. 3, we see this requires $k_{\mathrm{B}}T/E_{\mathrm{Th}} \simeq 0.48$ and $B \ll B_{\mathrm{c}}$), and its maximum value is

$$\langle ZT \rangle_{\max} \;\; \simeq \;\; 0.27 \times \frac{1}{N^2}, \tag{28}$$

while it will be much smaller for $k_{\mathrm{B}}T \ll E_{\mathrm{Th}}$ or $k_{\mathrm{B}}T \gg E_{\mathrm{Th}}$. For large magnetic fields, $B \gg B_{\mathrm{c}}$, the value of $\langle ZT \rangle$ at any $T$ is half that with $B = 0$ at the same $T$.

---

[4] We thank the anonymous referee for pointing out that this linear $T$ regime has a $N^{-3}$ dependence, which led us to discuss all four regimes in this manner.

# 3   Universality of the Seebeck coefficient and figure of merit

Mesoscopic conductance fluctuations are well-known to have a "universal" magnitude, and are hence referred to as *universal conductance fluctuations*. This magnitude is given by the square-root of the variance which goes like $N_{\rm L}^2 N_{\rm R}^2 / N^4$, where $N = N_{\rm L} + N_{\rm R}$. Hence, they are "universal" in the sense that they do not change if one multiplies the width of all leads by the same amount. To be more precise, if we define an asymmetry parameter

$$n_{\rm asym} = \frac{N_{\rm L} - N_{\rm R}}{N}, \tag{29}$$

then the variance of the conductance fluctuations can be written as

$$\mathrm{var}[G] = \left( \frac{e^2}{h} \left( 1 - n_{\rm asym}^2 \right) \right)^2 \times \left( \begin{array}{c} \text{dimensionless function} \\ \text{of } k_{\rm B} T / E_{\rm Th} \ \& \ B/B_c \end{array} \right),$$

which is universal in the sense that it does not depend on $N$. Actually, this is only true when $\mathrm{var}[G]$ does not depend on $E_{\rm Th}$ (i.e. at low enough temperature that $k_{\rm B} T \ll E_{\rm Th}$), because $E_{\rm Th}$ in Eq. (12) changes when one changes $N$ (by rescaling the lead widths), whether or not one rescales the system size by the same amount as the lead widths.

   Both the average figure of merit $\langle ZT \rangle$ in Eq. (27), and the typical magnitude of the Seebeck coefficient $\sqrt{\langle S^2 \rangle}$ in Eq. (23) depend on $N = N_{\rm L} + N_{\rm R}$, and so are not universal in the same way as the universal conductance fluctuations. However, they are universal in the sense that they are independent of the asymmetry parameter $n_{\rm asym}$. For example, the value of $\langle ZT \rangle$ or $\langle S^2 \rangle$ is the same for a system with highly asymmetric leads with $N_{\rm L} = 100$ and $N_{\rm R} = 300$, as it is for a system with symmetric leads with $N_{\rm L} = N_{\rm R} = 200$. Unlike the universality of conductance fluctuations, this universality of $\langle ZT \rangle$ and $\langle S^2 \rangle$ holds for all $T$, since $E_{\rm Th}$ in Eq. (12) depends on $N$ but not on $n_{\rm asym}$.

# 4   Full probability distributions

To calculate the full probability distribution of the thermoelectric coefficients, $S$ and $\Pi$, or the figure of merit, $ZT$, we calculate all moments of these quantities. For large $N$, we can use the same logic as above Eqs. (7) to write these moments as

$$\langle S^{2m} \rangle \ = \ \frac{\langle I_1^{2m} \rangle}{\langle I_0 \rangle^{2m}} \left[ 1 + \mathcal{O}[1/N] \right], \tag{30}$$

$$\langle (ZT)^m \rangle \ = \ \frac{\langle I_1^{2m} \rangle}{\langle I_2 \rangle^m \langle I_0 \rangle^m} \left[ 1 + \mathcal{O}[1/N] \right] \tag{31}$$

for integer $m$. It is also important to note that $\langle S^{2m} \rangle$ goes like $N^{-2}$, while $\langle S^{2m+1} \rangle$ goes like $N^{-3}$. Thus to leading order in $1/N$ we can take the odd moments of $S$ to be zero. Of course, the moments of the Peltier coefficients are simply given by the moments of the Seebeck coefficient via $\langle \Pi^m \rangle = T^m \langle S^m \rangle$ for even and odd $m$.

   Thus to calculate $\langle S^{2m} \rangle$ or $\langle (ZT)^m \rangle$, we need the average of $I_1^{2m}$ which contains the product of $2m$ transmission functions. This means that we need to calculate the average of arbitrary products of transmission functions. For this, it is convenient to use Eq. (10) to define $\delta \mathcal{T}(E)$ as the fluctuations of the transmission away from its average value, $\langle \mathcal{T}(E) \rangle$. As $\langle \mathcal{T}(E_i) \rangle$ is independent of $E_i$, see Eq. (9), nothing changes if one replaces $\mathcal{T}(E)$ by $\delta \mathcal{T}(E_i)$ in the integrand

of $I_1$. Thus

$$\langle I_1^{2m} \rangle = \int \frac{dE_1 dE_2 \cdots dE_{2m}}{h^m} E_1 f'(E_1) E_2 f'(E_2)) \cdots E_{2m} f'(E_{2m})$$
$$\times \langle \delta\mathcal{T}(E_1)\delta\mathcal{T}(E_2)\cdots\delta\mathcal{T}(E_{2m})\rangle, \tag{32}$$

where, without loss of generality, we have measured all energies from the electrochemical potential.

Then, the diagrammatics performed in section 5.3 gives us the averages of transmission functions in Eq. (32). For example, the average for $m = 4$ to leading order in $1/N$ is

$$\langle \delta\mathcal{T}(E_1)\delta\mathcal{T}(E_2)\delta\mathcal{T}(E_3)\delta\mathcal{T}(E_4)\rangle = \langle \delta\mathcal{T}(E_1)\delta\mathcal{T}(E_2)\rangle\langle\delta\mathcal{T}(E_3)\delta\mathcal{T}(E_4)\rangle$$
$$+\langle \delta\mathcal{T}(E_1)\delta\mathcal{T}(E_3)\rangle\langle\delta\mathcal{T}(E_2)\delta\mathcal{T}(E_4)\rangle$$
$$+\langle \delta\mathcal{T}(E_1)\delta\mathcal{T}(E_4)\rangle\langle\delta\mathcal{T}(E_2)\delta\mathcal{T}(E_3)\rangle, \tag{33}$$

where the three terms contain all possible pairwise combinations of of $E_1, E_2, E_3, E_4$. Turning now to arbitrary $m$, the diagrammatic rules tell us that $\langle \delta\mathcal{T}(E_1)\delta\mathcal{T}(E_2)\cdots\delta\mathcal{T}(E_{2m})\rangle$ (to leading order in $1/N$) is a sum with $(2m)!/(2^m m!)$ terms, in which each term contains the product of pairwise averages, and the sum is over all pairwise combinations of $E_1, E_2, \cdots E_{2m}$ (with the ordering in each pair being irrelevant).

In the context of $\langle I_1^{2m} \rangle$ in Eq. (32), these sums over pairwise combination of energies are greatly simplified by the fact that we integrate over all energies. As a result each of the $(2m)!/(2^m m!)$ terms in the sum gives the same integral, each of which equals $m$ products of the integral in $\langle I_1^2 \rangle$. Thus

$$\langle I_1^{2m} \rangle = \frac{(2m)!}{2^m m!} \langle I_1^2 \rangle^m, \tag{34}$$

for any integer $m$. For what follows, we also note that the same rules easily give $\langle I_1^{2m-1} \rangle = 0$ for integer $m$ (to leading order in $1/N$). Substituting Eq (34) into Eq. (30) then gives

$$\langle S^{2m} \rangle = \frac{(2m)!}{2^m m!} \langle S^2 \rangle^m, \qquad \langle S^{2m+1} \rangle = 0, \tag{35}$$

to leading order in $1/N$. Similarly, using Eq. (31), we see that the moments of the figure of merit are

$$\langle (ZT)^m \rangle = \frac{(2m)!}{2^m m!} \langle ZT \rangle^m, \tag{36}$$

to leading order in $1/N$.

Since these results give all moments of the probability distributions, one can find the probability distribution itself using the results in Appendix A. Then one sees that the probability distribution of $I_1$ is a Gaussian;

$$P(I_1) = \sqrt{\frac{1}{2\pi\langle I_1^2\rangle}} \exp\left[-\frac{I_1^2}{2\langle I_1^2\rangle}\right]. \tag{37}$$

## 4.1 Full distribution of the thermoelectric coefficients

Given Eqs. (4,37) we see that the Seebeck coefficient, $S$, has a Gaussian probability distribution,

$$P(S) = \sqrt{\frac{1}{2\pi\langle S^2\rangle}} \exp\left[-\frac{S^2}{2\langle S^2\rangle}\right]. \tag{38}$$

This Gaussian at large $N$ was predicted from general arguments in Refs. [8,9], without giving the form of $\langle S^2 \rangle$ outside the limit of small $T$. Our Eq. (23) gives $\langle S^2 \rangle$ for arbitrary $T$ and $B$, and hence completely determines the distribution of $S$ for large $N$. Turning to the Peltier coefficient, the fact that $\Pi = TS$ in these systems directly means that $P(\Pi) = (2\pi \langle \Pi^2 \rangle)^{-1/2} \exp\left[-\Pi^2 / 2\langle \Pi^2 \rangle\right]$.

One can compare the Gaussian distribution of $S$ at large $N$, with the distributions at small $N$ in Refs. [8,9], they show discontinuities at small $S$ which are absent in the distribution at large $N$.

## 4.2 Full distribution of the figure of merit

Since $I_1$ follows a Gaussian distribution and $ZT$ goes like $I_1^2$, it only takes one line of algebra (see appendix A), to see that the distribution of $ZT$ is

$$
P(ZT > 0) \;=\; \sqrt{\frac{1}{2\pi \langle ZT \rangle ZT}} \; \exp\left[-\frac{ZT}{2\langle ZT \rangle}\right],
\tag{39}
$$

while $P(ZT < 0) = 0$. This probability distribution is full determined by its average, $\langle ZT \rangle$, given in Eq. (27).

This distribution tells us that the probability a sample has a value of $ZT$ which is more than $\alpha$ times the average, is given by $1 - \mathrm{erf}(\sqrt{\alpha/2})$, where $\mathrm{erf}(x)$ is an error function. It tells us that 2.5% of samples will have a value of $ZT$ which is more than 5 times bigger than the average, but only in 0.16% of samples will have a value of $ZT$ which is more than 10 times bigger than the average. The square-root divergence at small $ZT$ means there is a high probability of samples having smaller $ZT$ than average, for example there is a 25% chance that a sample has $ZT$ less than one tenth of the average.

## 5 Diagrammatics for open chaotic or disordered dots

This work uses the diagrammatic method developed in Refs. [28–33,35,36] for correlations in transport, both quantum noise effects and conductance fluctuations, which built on the earlier work on mesoscopic corrections [37–42]. These works showed that open quantum chaotic systems obey random matrix theory, as was already shown for closed systems in Refs. [26,27, 43], however in the process they developed simple diagrammatic rules to calculate averages of transmission functions which work for both random matrices and the semiclassical limit of quantum chaotic systems.

To draw the diagrams, one notes that the transmission function $\mathcal{T}(E) = \sum_{ij} t_{ij}(E) t_{ij}^*(E)$, where $j$ is summed over all modes of the left lead, and $i$ is summed over all modes of the right lead. Here $t_{ij}(E)$ is the $ij$th element of the part of the system's scattering matrix which corresponds to particles going from the left reservoir the the right reservoir. Each factor of $t_{ij}(E)$ or $t_{ij}^*(E)$ can be represented as a sum over all trajectories from left to right, so $\mathcal{T}(E_1)\mathcal{T}(E_2) = \sum_{i_1 j_1} \sum_{i_2 j_2} t_{i_1 j_1}(E_1) t_{i_1 j_1}^*(E_1) t_{i_2 j_2}(E_2) t_{i_2 j_2}^*(E_2)$, corresponds to four trajectories from left to right. It is convenient to draw the trajectories associated with factors of $t(E)$ as solid lines and the trajectories associated with factors of $t^*(E)$ as dashed line, and to use different colours for different energies. See the example in Fig. 4. Any solid trajectory that is not paired with a dashed trajectory (and vice-versa) at every moment will average to zero. Broadly speaking, this is because each trajectory's contribution to $t_{ij}(E)$ is proportional to a factor whose phase is that trajectory's action in units of $\hbar$. If a trajectory is unpaired, its phase will be statistically independent of the phase of the other trajectories, and averaging over the

(a) Example of physical trajectories         (b) Equivalent diagram

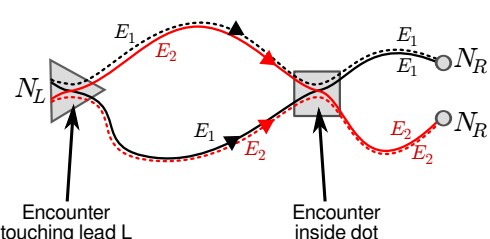

Figure 4: This sketch indicates how to go from (a) physical trajectories in an disordered open dot to (b) the slightly more abstract diagrams used to calculate the transport properties, such as the diagrams in Figs. 5 and 6. The equivalent for chaotic systems are trajectories which bounce back and forth inside the dot, but when unfolded look similar to the diagrams in (b), with paths diverging and converging from each other. In (b) we use a triangle to indicate an encounter that touches a lead, and a rectangle to indicate an encounter inside the system; this notation is carried over into Figs. 5 and 6.

ensemble will correspond to averaging over large fluctuations in the phase, which means the average will be zero. The simplest trajectory pairing is to pair the two trajectories with the same energy. This gives $\langle \mathcal{T}(E_1)\mathcal{T}(E_2)\rangle = \langle \mathcal{T}(E_1)\rangle\langle \mathcal{T}(E_2)\rangle$. Given Eq. (10), such pairings give no contribution to $\langle \delta\mathcal{T}(E_1)\delta\mathcal{T}(E_2)\rangle$, which means that the contributions to $\langle \delta\mathcal{T}(E_1)\delta\mathcal{T}(E_2)\rangle$ come from more complicated pairings. Such more complicated pairings involve a solid trajectory paired with a given dashed trajectory for part of the time, but pairs meet at "encounters" where the pairings swap, for examples see Figs. 4-6. There are many such contributions, and only the ones at leading order in $1/N$ are discussed here.

The diagrammatic rules are derived in sections VI and VII of Ref. [28], and are summarized in Ref. [44] (although Ref. [44] also includes rules for superconducting reservoirs). The rules relevant here are as follows.

1. A trajectory-pair consisting of one solid trajectory with energy $E_i$ and one dashed trajectory with energy $E_j$ gives a factor of

$$\frac{1}{N}\frac{1}{1 - \mathrm{i}(E_i - E_j)/E_{\mathrm{Th}} + \chi\,(B/B_{\mathrm{c}})^2}, \tag{40}$$

with $\chi = 1$ for time-reversed trajectories (marked with TR in Fig. 6) and with $\chi = 0$ otherwise.

2. A trajectory-pair that ends at lead $i$ while not in an encounter, gives a factor of $N_i$. In Figs. 5 and 6, these are marked by the small circles on the left of the diagram for lead L, and small circles on the right for lead R.

3. An encounter touching lead $i$ gives a factor of $N_i$, irrespective of how many trajectory-pairs meet at the encounter, and thereby end at the lead. In Fig. 5 these are marked by the triangles on the left of the diagram for lead L and triangles on the right of the diagram for lead R (there are no such contributions in Figs. 6). This rule is not general, but applies to all the encounters touching leads that occur in the diagrams shown in this work.

4. An encounter inside the dot gives a factor of $-N \times (1 + \kappa)$. In Figs. 5 and 6, these are marked by the rectangular boxes. The factor of $(-N)$ is there for any encounter, no

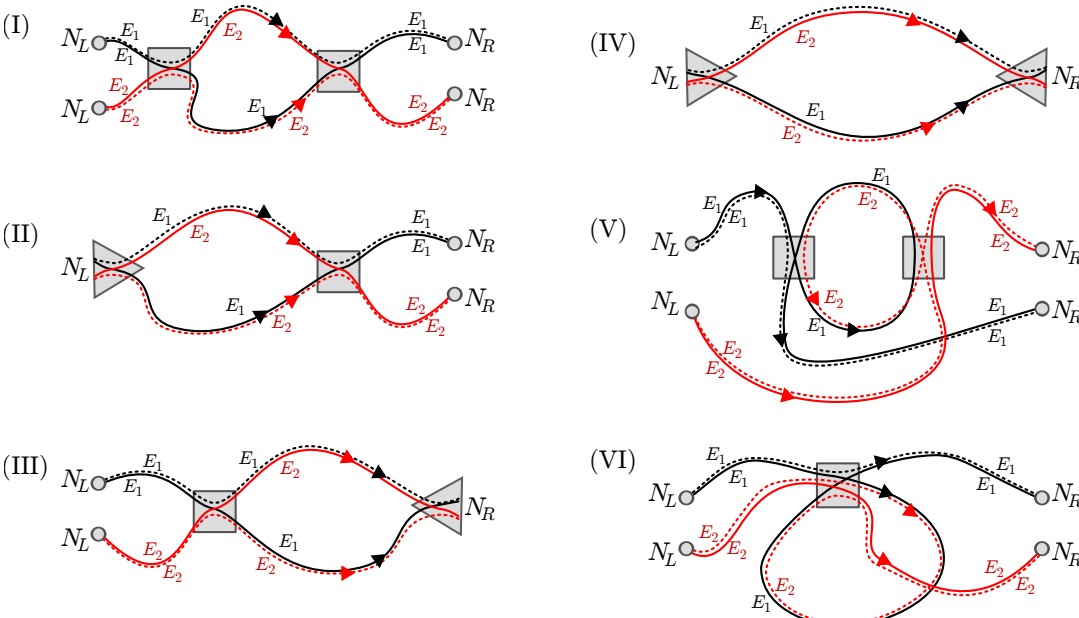

Figure 5: The leading order in $1/N$ contributions to $\left\langle \delta\mathcal{T}(E_1)\delta\mathcal{T}(E_2)\right\rangle$ without time-reversed trajectory-pairs; in other words, the trajectories in these contributions all go in the same direction in all trajectory pairs. These contributions are hence independent of the external $B$-field. The physical meaning of such contributions is indicated in Fig. 4. The solid black trajectory correspond to a contribution to $t(E_1)$, and the dashed black trajectory corresponds to contribution to $t^*(E_1)$. The solid red trajectory correspond to a contribution to $t(E_2)$, and the dashed red one to contribution to $t^*(E_2)$. Triangles indicating encounters that touch the leads, and rectangles indicating encounters inside the system. Note that contributions II and III are only different because of how they couple to the leads. Contribution II has the four paths together in an encounter at lead L, while they are in two pairs at lead R. In contrast, contribution III has the paths in two pairs at lead l, while the encounter is at lead R. One must have both these contributions.

matter how many trajectories meet at the encounter. The $N$-independent factor of $\kappa$ contains the encounter's dependence on energy differences and B field, and depends on details of the encounter in the manner shown in Fig. 7. Note that $\kappa = 0$ for any encounter in the limit where all energies are equal ($E_1 = E_2$, etc) and there is no external magnetic field ($B = 0$).

## 5.1 Diagrams for $\left\langle T(E)\right\rangle$

At leading order in $1/N$ there is only one diagram for $\left\langle T(E)\right\rangle$, it is one in which the trajectories form a single pair from lead L to lead R. Thus there is one trajectory-pair and it is not time-reversed, thus its weight is given by rule 1 in the above list, with $E_i = E_j = E$ and $\chi = 0$. One end of the trajectory pair is on lead L and the other on lead R, with weights given by rule 2 in the above list. There are no encounters, so the other rules in the above list are not necessary. Thus we conclude that $\left\langle T(E)\right\rangle$ to leading order in $1/N$ is given by the result in Eq. (9).

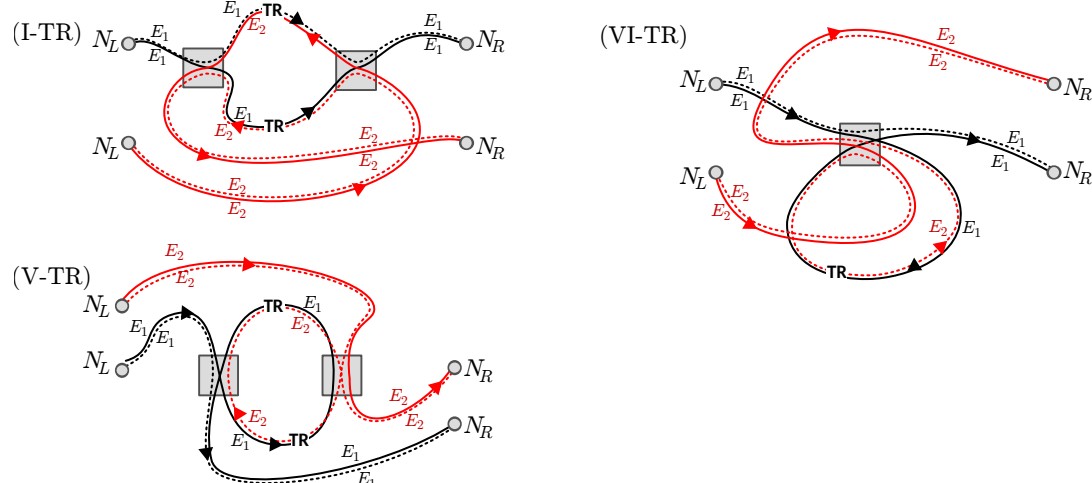

Figure 6: Leading order in $1/N$ contributions to $\langle \delta \mathcal{T}(E_1)\delta \mathcal{T}(E_2)\rangle$ which contain some time-reversed (TR) trajectory-pairs (some of the trajectory pairs consist of trajectories going in opposite directions). These time-reversed trajectory-pairs are marked with "TR", such trajectory-pairs are cooperons in the language of disordered systems. Each diagram here corresponds on one in Fig. 5, but with the red trajectories passing though the encounters in the opposite way. However, this is not possible for the contributions in Fig. 5 with encounters at the leads (contributions II, III, and IV), meaning there are only the above three diagrams containing TR trajectory-pairs.

## 5.2 Diagrams for $\langle \delta \mathcal{T}(E_1)\delta \mathcal{T}(E_2)\rangle$

To calculate the typical magnitude of the thermoelectric coefficients or the average figure of merit in the large $N$ limit, we need to evaluate all the leading-order diagrams that contribute to $\langle \delta \mathcal{T}(E_1)\delta \mathcal{T}(E_2)\rangle$ to arrive at Eq. (11). These were discussed in Ref. [28,33], and are shown in Fig. 5. Using the diagrammatic rules, one can convince oneself that these are the only diagrams that are order zero in $N$, and that all others (such as those including weak-localization loops) are higher order in $1/N$. Thus, for large $N$, we only need to consider the diagrams in Figs. 5 and 6.

Giving weights to these diagrams following the above rules, we see that contribution I in Fig 5 has two encounters with $\kappa = 0$, so its total weight is

$$W_{\mathrm{I}} = \frac{N_{\mathrm{L}}^2 N_{\mathrm{R}}^2}{N^4}\left[1 + \frac{(E_2 - E_1)^2}{E_{\mathrm{Th}}^2}\right]^{-1}. \tag{41}$$

The weights of contributions II, III and IV are

$$W_{\mathrm{II}} = -\frac{N_{\mathrm{L}} N_{\mathrm{R}}^2}{N^3}\left[1 + \frac{(E_2 - E_1)^2}{E_{\mathrm{Th}}^2}\right]^{-1}, \tag{42}$$

$$W_{\mathrm{III}} = -\frac{N_{\mathrm{L}}^2 N_{\mathrm{R}}}{N^3}\left[1 + \frac{(E_2 - E_1)^2}{E_{\mathrm{Th}}^2}\right]^{-1}, \tag{43}$$

$$W_{\mathrm{IV}} = \frac{N_{\mathrm{L}} N_{\mathrm{R}}}{N^2}\left[1 + \frac{(E_2 - E_1)^2}{E_{\mathrm{Th}}^2}\right]^{-1}, \tag{44}$$

which means that they sum to zero; $W_{\mathrm{II}} + W_{\mathrm{III}} + W_{\mathrm{IV}} = 0$.

Contributions V and VI have more complicated encounters with non-zero $\kappa$. Contribution V has two encounters, with trajectories pairs contributing the same as in $W_{\mathrm{I}}$, however

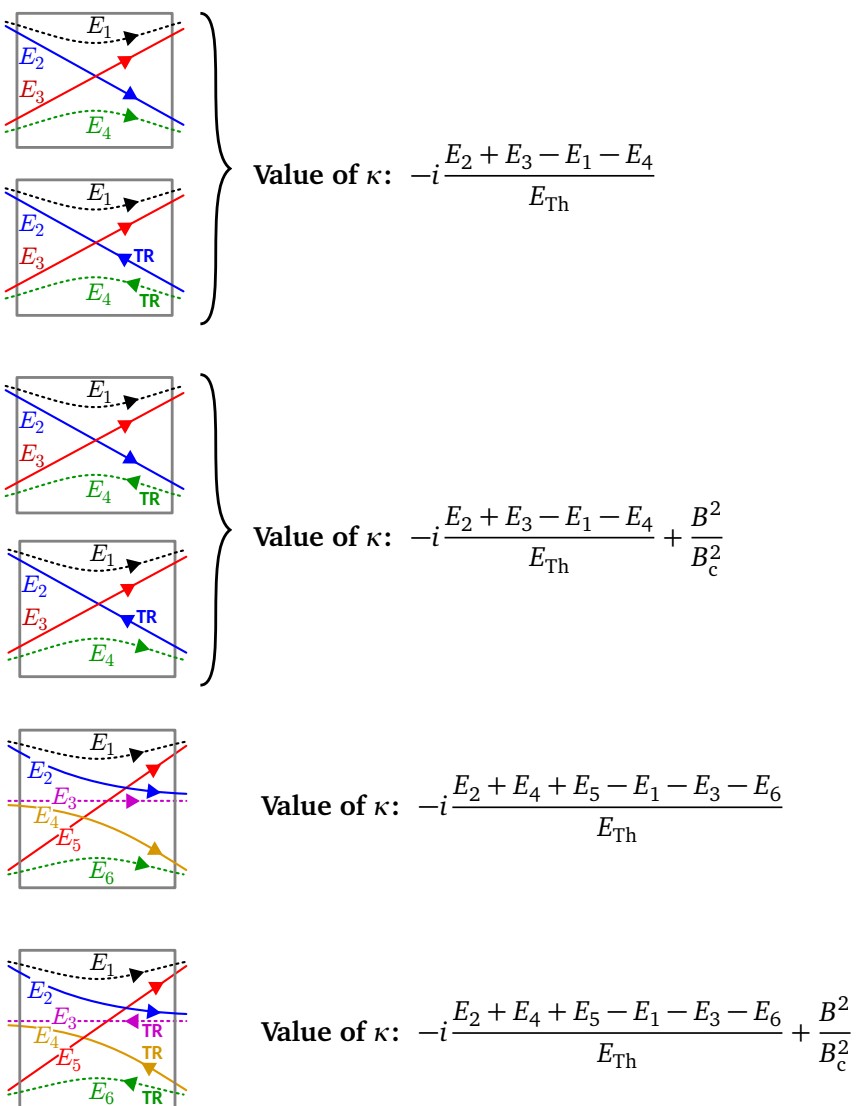

Figure 7: An encounter's contribution is $-N \times [1 + \kappa]$, where $\kappa$'s dependence on energies and the magnetic field is different for different encounters. The value of $\kappa$ for various encounters is given here. Sections VI and VII of Ref. [28] explain the origin of these contributions.

one encounter has $\kappa = i(E_2 - E_1)/E_{\mathrm{Th}}$, and the other has $\kappa = -i(E_2 - E_1)/E_{\mathrm{Th}}$. Hence, the contribution's weight is

$$W_{\mathrm{V}} \quad = \quad \frac{N_{\mathrm{L}}^2 N_{\mathrm{R}}^2}{N^4}, \tag{45}$$

where the energy dependence that came from the encounters exactly cancelled the energy dependence that came from the trajectory-pairs. The contribution VI has an encounter with $\kappa = i(E_2 - E_1)/E_{\mathrm{Th}}$, hence the contribution's weight is

$$W_{\mathrm{VI}} \quad = \quad -\frac{N_{\mathrm{L}}^2 N_{\mathrm{R}}^2}{N^4}, \tag{46}$$

where again there is exact cancellation between the energy dependence that came from the encounter and that which came from the trajectory-pair. Hence, these two contributions also sum to zero; $W_{\mathrm{V}} + W_{\mathrm{VI}} = 0$.

Now we turn to the contributions which involve some time-reversed trajectory-pairs, which are shown in Fig. 6. The contribution I-TR has encounters with the same weight as contribution I, but with the trajectory-pairs marked with "TR" have extra factors of $(B/B_c)^2$ in the denominator, hence its weight is

$$W_{\text{I}-\text{TR}} \;\; = \;\; \frac{N_{\text{L}}^2 N_{\text{R}}^2}{N^4}\left[1 + \frac{(E_2 - E_1)^2}{E_{\text{Th}}^2} + \frac{B^2}{B_{\text{c}}^2}\right]^{-1}. \tag{47}$$

The contributions V-TR and VI-TR are similar to those of V and VI, but both the encounters and the trajectory pairs have extra factors of $(B/B_c)^2$. Remarkably, these factors are arranged in such a manner that they all cancel, and the contributions are $B$-independent. Thus

$$W_{\text{V}-\text{TR}} \;\; = \;\; -W_{\text{IV}-\text{TR}} = \frac{N_{\text{L}}^2 N_{\text{R}}^2}{N^4}, \tag{48}$$

so again these two contributions sum to zero. Note that it would be very odd if contributions V-TR and VI-TR did not sum to zero, since one expects on general grounds that the correlations decay with increasing $B$, so non-cancelling $B$-independent contributions should not exist.

Thus, in conclusion, the sum of all contributions to $\left\langle \delta \mathcal{T}(E_1) \delta \mathcal{T}(E_2) \right\rangle$ is equal to the sum of the weights of contributions I and I-TR, and this is what is given in Eq. (11). All the above cancellations between contributions makes one wonder if there is a deeper principle at play here, however we see no such principle at the level of the sums over semiclassical trajectories considered here.

## 5.3 Diagrams for higher order correlators

The objective of this section is to argue that the diagrammatic rules tell us that an $M$th order transmission correlators of the form $\left\langle \delta \mathcal{T}(E_1) \delta \mathcal{T}(E_2) \cdots \delta \mathcal{T}(E_M) \right\rangle$, is given by all possible pairwise groupings of the transmission coefficients, see in and below Eq. (33).

The $M$th order transmission correlators of the form $\left\langle \delta \mathcal{T}(E_1) \delta \mathcal{T}(E_2) \cdots \delta \mathcal{T}(E_M) \right\rangle$, consists of $2M$ trajectories from lead L to lead R, as sketched on the left in Fig. 8. Such contributions average to zero unless all the trajectories they contain are paired up at all time (switching pairing at encounters). In addition, because of the definition in Eq. (10), a contribution is also zero if any trajectory is only paired with its partner of the same energy.

We will see that for large $N$, the leading order contributions to this correlation are of order $N^0$, with corrections going like $1/N$. With this in mind, our objective is to find all ways of pairing the $2M$ trajectories in manners that give a contribution at order $N^0$.

We start by taking the first pair of trajectories (those with energy $E_1$) and connecting them with the $i$th pair of trajectories (those with energy $E_i$), such that they meet at encounters and exchange pairings. This connection can be via any of the diagrams in Figs. 5-6, all of which are order $N^0$. If we now take the $j$th pair of trajectories (those with energy $E_j$, and try to connect them (via encounters) with trajectories of pair 1 or pair $i$, we find we always get a diagram which is of order $1/N$ or smaller. Thus to get a contribution of order $N^0$, we must connect the $j$th pair with another pair which has not yet been involved in any encounters (i.e. the pair with energy $E_k$ for $k \neq 1, i, j$). Trajectories that are not connected are statistically independent, and so can be averaged separately. Thus order $N^0$ contributions are those where each trajectory pair with exactly one other, all other contributions are smaller, and so can be neglected. The result is that for $M = 4$, we get Eq. (33), and for even $M$ (so $M = 2m$ with integer $m$) we get the result outlined below Eq. (33). In contrast for odd $M$, there is an odd number of trajectory pairs, and so one cannot connect each pair with one other (there will always be a pair left over). Thus $\left\langle \delta \mathcal{T}(E_1) \delta \mathcal{T}(E_2) \cdots \delta \mathcal{T}(E_M) \right\rangle$ is of order $1/N$ for odd $M$, which means we can take it to be zero when working at order $N^0$.

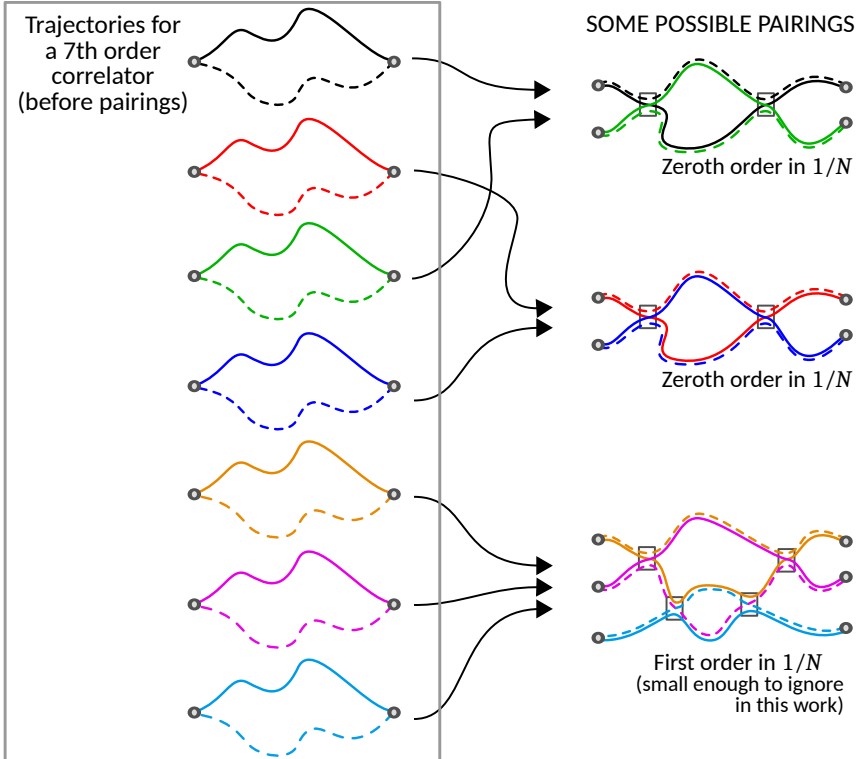

Figure 8: Contributions to $\left\langle \delta\mathcal{T}(E_1)\delta\mathcal{T}(E_2)\cdots\delta\mathcal{T}(E_M)\right\rangle$ for $M = 7$. On the left, each $\delta\mathcal{T}(E_i)$ is represented by two trajectories of the same colour joined at beginning and end. Only contributions where the trajectories are connected contribute to the average. On the right, we show one possible way of connecting them. All trajectories are in pairs, but they swap pairings at the encounters (black rectangles). If any colour is paired with itself alone, then its contribution is zero. The full contribution is the product of the individual connected contributions. The connected contributions involving only two pairs of trajectories (such as black-green or red-blue here) are zeroth order in $1/N$, Those involving three or more pairs of trajectories are higher order in $1/N$; for example the connected contribution involving three colours (orange-purple-cyan) is of order $/N$. Thus the product over all connected contributions will be of order $N^0$, if all connected contributions involve only two pairs of trajectories (i.e. two colours). See section 5.3 for more details.

## 6 Conclusions

Here, we have provided a quantitative theory for the thermoelectric response and the figure of merit of large open quantum dots. The dots may be disordered or chaotic, and are coupled to two leads, left (L) and right (R), which carry a large number of modes, $N \gg 1$. Such systems have long been known to have a rather weak thermoelectric response, however to the best of our knowledge no quantitative results existed outside the regime of very low temperatures. Quantitative theories such as this are necessary, if one wishes to use the thermoelectric response as a probe nanostructures.

    We show that the thermoelectric response of such systems is always small, but is peaked when the temperature is very close to half the Thouless energy ($k_{\mathrm{B}}T/E_{\mathrm{Th}} \simeq 0.48$), and when the external magnetic flux through the dot is zero. This condition maximises both the typical Seebeck and Peltier coefficients , $S_{\mathrm{typical}}$ and $\Pi_{\mathrm{typical}}$, and the average thermoelectric figure of merit, $\langle ZT \rangle$. These quantities are also found to exhibits a type of universality, in the sense that

it does not depend on the asymmetry between the number of modes in lead L and R ($N_L - N_R$), but only on the sum of the two, $N = (N_L + N_R)$.

## Acknowledgements

R.W. thanks the QuEnG project of the Université Grenoble Alpes, which is part of the French ANR-15-IDEX-02. K.S. was supported by JSPS Grants-in-Aid for Scientific Research (JP16H02211 and JP17K05587).

## A  From moments to probability distribution

In section 4, we have a probability distribution, $P(y)$, whose moments obey

$$\langle y^{2m-1} \rangle = 0, \quad \text{and} \quad \langle y^{2m} \rangle = \frac{(2m)!}{2^m m!} \langle y^2 \rangle^m, \tag{49}$$

for integer $m$, where $\langle \cdots \rangle = \int dy\, P(y)(\cdots)$. For a Gaussian probability distribution,

$$P(y) = \sqrt{\frac{\alpha}{\pi}} \exp\left[-\alpha y^2\right], \tag{50}$$

with $\alpha = 1/\left(2\langle y^2 \rangle\right)$, one can calculate the moments, and see that they obey Eq. (49).

If one wishes to prove this in a deductive manner, rather than via the above observation, one can start by defining the generating function

$$F(t) \equiv \langle e^{iyt} \rangle = \sum_{n=0}^{\infty} \frac{it\langle y^n \rangle}{n!}. \tag{51}$$

This means that $F(t) = \int dy\, e^{iyt} P(y)$, and hence the probability distribution, $P(y)$, can be found from $F(t)$ via the Fourier transform,

$$P(y) = \int \frac{dt}{2\pi} e^{-iyt} F(t). \tag{52}$$

Thus, once one has $F(t)$, one can find $P(y)$.

In the case of interest here, substituting Eq. (49) into Eq. (51), we define $n = 2m$, and find that

$$F(t) = \sum_{m=0}^{\infty} \frac{\left(-t^2\langle y^2 \rangle\right)^m}{2^m\, m!} = \exp\left[-\tfrac{1}{2}\langle y^2 \rangle t^2\right]. \tag{53}$$

Thus, the Fourier transform gives

$$P(y) = \sqrt{\frac{1}{2\pi\langle y^2 \rangle}} \exp\left[-\frac{y^2}{2\langle y^2 \rangle}\right]. \tag{54}$$

Next we wish to consider the probability distribution $P(z)$, whose moments obey

$$\langle z^m \rangle = \frac{(2m)!}{2^m m!} \langle z \rangle^m. \tag{55}$$

By comparison with Eq. (49) we see that it is the same with $z = y^2$, and we can use this fact to get $P(z)$ from Eq. (54). We have $P(z)dz = 2P(y)dy$, where the factor of two is due to contributions to $P(z)$ from both positive and negative $y$, hence

$$P(z) = 2P(y)\Big|_{y=\sqrt{z}} \frac{d}{dz}\sqrt{z} = \sqrt{\frac{1}{2\pi\langle z\rangle z}} \exp\left[-\frac{z}{2\langle z\rangle}\right], \qquad (56)$$

which is an exponential decay with square-root divergence at small $z$.

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
