# Peer review of "Thermoelectric coefficients and the figure of merit for large open quantum dots"

_SciPost Physics, doi:SciPost Phys. 6, 012 (2019)_

## Round 2 · Referee Report · Anonymous (Referee 1) · 2018-9-12

Strengths

  1. The paper quantitatively addresses the problem of weak thermoelectric response of large open quantum dots which was left open by reviewed prior works.

  2. The averages of thermoelectric quantities, their powers and even their full statistics are explicitly derived. The used diagrammatic technique is very helpful in understanding the derivation, in particular the nontrivial factorization into pair contributions (Fig. 6).

  3. The results indicate that a formula indicated in previous work without derivation (Ref. [10]) although correct is of quite limited relevance. Instead, the relevant temperature dependence is almost completely governed by higher order terms in $k T/ E_{Th}$ computed in this paper.

4 The paper is clearly written and good to follow, the figures are very good and the appendix is helpful.

Weaknesses

None

Report

This is an nicely written and interesting work and I recommend its publication. Apart from the unproblematic changes requested below I have two optional questions /suggestions:

1.
The cancellations that causes only diagram I and I-TR to contribute make me wonder if there is a simple deeper principle causing this. Do the authors have an idea / suggestion?

2.
The authors are careful to mention that all discussed effects are small and emphasize thermoelectrics as a probe for the physics of dots. Can they comment briefly in the outlook on electron-electron interactions perhaps with some pertinent reference? I guess these break the symmetry that causes $\langle S \rangle =0$? Is their any prospect of including these with their / a similar approach?

Requested changes

Warnings issued while processing user-supplied markup:

  • Inconsistency: plain/Markdown and reStructuredText syntaxes are mixed. Markdown will be used.
    Add "#coerce:reST" or "#coerce:plain" as the first line of your text to force reStructuredText or no markup.
    You may also contact the helpdesk if the formatting is incorrect and you are unable to edit your text.

1. Abstract. The "asymmetry between the left and right leads" is easily misunderstood in the abstract for the standard multipicative asymmetry. I suggest some modification to distinguish this: "...does not depend on the additive asymmetry between the left and right leads."

2. Eq. (19): the paper stresses that the previously reported result (19) is essentially irrelevant since it holds only for very small numerical ratios of $T/E_{Th}$. It is not so clear what the simple behavior at low T now is, in particular the N dependence. Looking at Fig. 2, it seems to me that a rough linear interpolation between the origin and the maximum is the relevant result, i.e.,

$\langle S^2\rangle \propto \langle S^2\rangle_{max} (kT/E_Th)$

which at scales as $1/N^3$ rather than $1/N^4$ or $1/N^2$. It is only when fixing an $N$-dependent temperature $kT \sim E_{Th} \sim N \Delta$ that one gets $\langle S^2 \rangle_{max} \propto 1/N^2$ as in Eq. (20). Please comment on this. I would even suggest to point out more clearly in the introduction and conclusions hat the result of Ref. [10] in some way puts one on the wrong track. This does not become clear until p. 6. and actually is key insight of this work in my view.

  1. After Eq. (20): It may be worth to comment on the $T$-position of the maximum of (17): it seems to be the same for low and high fields and but in between makes a (small?) excursion for finite fields $B \sim B_c$?

  1. After Eq. (21): I was surprised the authors did not comment on the fact that their result (21) is up to constant factors just Eq. (17):

$\langle ZT \rangle = (3/\pi^2) (e/k_B)^2 \langle S^2 \rangle = \langle S^2 \rangle / L $.

where L is the Lorenz number. The constant factor derives from $\langle I_2\rangle /\langle I_0\rangle e^2 T^2$ by Eq. (7-8). I was not sure if Eq. (21) is now an unexpected result given (17) or should one expect it (does Wiedemann-Franz apply)? Please clarify.

5. On p. 7: "because the leading term in the expansion of .... is of order $x^2$" I guess the leading correction (next-to-leading term) is meant here, the leading term is 1/b.

-----6. On p. 8, Eq. (10): the $lim_{T\to 0}$ is confusing here (the limit is just zero), write $k_B T \ll E_{Th}$ instead.

-----7. On p. 11: "it is only take one line of algebra" needs a fix.

-----8. On p. 12, Fig. 3: the rules mention "an encounter inside the dot". I cannot make out from the diagrams why lines I-IV do not run through the dot ($\kappa=0$) and V-VI do run through it ($\kappa \neq 0$). Where is the dot to be seen ? It is not stated in the caption where the dot is to be seen and that the boxes are the "encounters".

-----9. On p. 13: Fig. 5 is referenced here but it appears only on p. 16 after unrelated text. Move Fig. 5 to p. 14.

-----10. On p. 17: "those with E_J" should read "with E_j" ?

-----11. On p. 17: Summary: "We have provided" ... "We showed ..." perhaps? Fix: "have long been knowN to...", "thermoeleCTRIC figure of merit"

---

## Round 3 · Referee Report · Anonymous · 2018-12-6

Report

I'm fully content with the changes made by the authors and their answers to my optional questions.

I maintain my previous list of strengths of the paper and recommend its publication in the present form.

---

## Round 3 · Author Response

We thank the referee for a very thorough reading of this manuscript, and for their insightful recommendation about the results and their presentation. We have implemented all the changes that the referee suggested; we explain these changes point by point below.

We added new figures 2 and 4, to help the reader's comprehension. We have also moved an intuitive explanation of our results from the body of the text into the introduction (the new section 1.1); this explains why such systems always have small thermoelectric effects, and allows one to guess their approximate magnitude.

DETAILED RESPONSE TO THE MAIN QUESTIONS IN THE REPORT

1) The referee raises a very interesting point, when he asks why there is so much cancellation between contributions. It is possible that there is a deeper principle at play here, but we do not know what. We see no such principle at the level of the sums over semiclassical trajectories considered here. However, it may be that treating random matrices as formal mathematical objects (without the crutch of the intuition gained from trajectories traversing a real chaotic or disordered system) would lead to such deeper principles.

2) Electron-electron interactions are beyond the scope of the semiclassical single-particle method considered here, and in general much less in known about such semiclassics in the presence of such interactions. The problem of interactions + disorder is one of the toughest ones in theoretical physics, and we know of few works on the thermoelectric responses of systems with interactions and disorder. We have added some sentences to Section 1.2 to mention two works in this direction in similar (large N) mesoscopic structures. + P.M. Chaikin and G. Beni, Thermopower in the correlated hopping regime, Phys. Rev. B 13, 647 (1976). + J. W. P. Hsu, A. Kapitulnik, and M. Yu. Reizer, Effect of electron-electron interaction on the thermoelectric power in disordered metallic systems, Phys. Rev. B 40, 7513 (1989). However, we mention that it is rather easy to break <S>=0, even without electron-electron interactions, by considering systems with N ~ 1, so the conduction is of order (or less than) a single channel.
Our recent review contains many examples of such systems; both non-interacting (mostly sections 4-6), and systems with electron-electron and electron-phonon interactions (mostly sections 7-9).

---

## Round 3 · List of Changes

RESPONSE TO REQUESTED CHANGES

1) We worry that the words "additive asymmetry" will not mean much to readers,
so we decided to be more explicit in the abstract. Saying
"... they do not depend on the asymmetry between the left and right leads (N_L − N_R),
even though they depend on (N_L + N_R)."
We also modified section 3 "Universality of the Seebeck coefficient and figure of merit" to clarify this point.

2) The dependence <S^2> on N is non-linear in both T and N, because as the referee points out E_{Th} goes like N, and <S^2> is a non-linear function of E_{Th}. We have added a discussion about the 4 regimes
i) very small T where it goes like N^{-4}T^2 as in Ref [10]
ii) the referee's regime where it goes like N^{-3}T
iii) the peak where it goes like N^{-2}
iv) high temperatures where it goes like (NT)^{-1}
and cite the (anonymous) referee for suggesting regime (ii).

3) The referee is correct that the peak makes a small excursion for finite fields. We have added a plot to fig 3 which shows how the peak moves from x=0.48 at b=1 up to nearly x=0.6 at b~2 before dropping back to x=0.48 at b=infinity, see fig 3b.

4) The referee is 100% correct, in this regime of weak energy dependence the ratio G/K is the Lorentz ratio (with corrections of order 1/N discussed in Ref [10]). We have added a new section 2.2 about the Wiedemann-Franz ratio.

5) The referee is correct, there was a typo, we now refer to the "next-to-leading" term, not the leading term.
We have taken the opportunity to calculate the prefactor on this term, now given in a new Eq. (17).

6) We have made the correction that the referee suggested.

7) We have fixed this typo.

8) The referee writes "On p. 12, Fig. 3: the rules mention an encounter inside the dot. I cannot make out from the diagrams why lines I-IV do not run through the dot (kappa=0) and V-VI do run through it (kappa neq 0). Where is the dot to be seen? It is not stated in the caption where the dot is to be seen and that the boxes are the encounters."

We have clarified the connection between physical trajectories in a dot, and the slightly abstract diagrams, in a new Fig.4. We have also modified the diagrams to clarify which encounters are touching the leads, and which are inside the dot, the former are now marked by a triangular box and the latter by a rectangular box.
The value of kappa for the encounters marked by rectangular boxes are then given in Fig. 7, and depend on the energies of the trajectories entering the encounter, and the magnetic field.

9) We have done our best to put the figures as close as possible to the place where they are cited in the text.
10) We have fixed this typo.
11) We have fixed these typos in the conclusions.

---

## Editorial Decision

published